# Cytokine-Induced iNOS in A549 Alveolar Epithelial Cells: A Potential Role in COVID-19 Lung Pathology

**DOI:** 10.3390/biomedicines11102699

**Published:** 2023-10-03

**Authors:** Amelia Barilli, Giulia Recchia Luciani, Rossana Visigalli, Roberto Sala, Maurizio Soli, Valeria Dall’Asta, Bianca Maria Rotoli

**Affiliations:** 1Laboratory of General Pathology, Department of Medicine and Surgery, University of Parma, 43125 Parma, Italy; amelia.barilli@unipr.it (A.B.);; 2Immunohematology and Transfusion Medicine, University Hospital of Parma, 43125 Parma, Italy

**Keywords:** baricitinib, IFNγ, IL-1β, iNOS, nitric oxide, TNFα

## Abstract

Background. In COVID-19, an uncontrolled inflammatory response might worsen lung damage, leading to acute respiratory distress syndrome (ARDS). Recent evidence points to the induction of inducible nitric oxide synthase (*NOS2*/iNOS) as a component of inflammatory response since *NOS2* is upregulated in critical COVID-19 patients. Here, we explore the mechanisms underlying the modulation of iNOS expression in human alveolar cells. Methods. A549 WT and IRF1 KO cells were exposed to a conditioned medium of macrophages treated with SARS-CoV-2 spike S1. Additionally, the effect of IFNγ, IL-1β, IL-6, and TNFα, either alone or combined, was addressed. iNOS expression was assessed with RT-qPCR and Western blot. The effect of baricitinib and CAPE, inhibitors of JAK/STAT and NF-kB, respectively, was also investigated. Results. Treatment with a conditioned medium caused a marked induction of iNOS in A549 WT and a weak stimulation in IRF1 KO. IFNγ induced *NOS2* and synergistically cooperated with IL-1β and TNFα. The inhibitory pattern of baricitinb and CAPE indicates that cytokines activate both IRF1 and NF-κB through the JAK/STAT1 pathway. Conclusions. Cytokines secreted by S1-activated macrophages markedly induce iNOS, whose expression is suppressed by baricitinib. Our findings sustain the therapeutic efficacy of this drug in COVID-19 since, besides limiting the cytokine storm, it also prevents *NOS2* induction.

## 1. Introduction

SARS-CoV-2 infection causes a broad spectrum of clinical symptoms, ranging from asymptomatic infection to critical illness, which can rapidly progress to severe complications and death. The lungs are the principal target of viral infection and represent the main pathological site of severe COVID-19.

Inflammatory cytokines are essential players in the orchestration of an immune response to viral pathogens; however, uncontrolled activation of immune cells can result in an exacerbated inflammatory response that can ultimately worsen lung injury and lead to acute respiratory distress syndrome (ARDS), systemic inflammatory response syndrome (SIRS) and multiple organ failure [1,2,3,4,5]. ARDS is the most severe form of lung injury and is characterized by diffuse alveolar damage and increased endothelial permeability, leading to edema and respiratory insufficiency [6]. SIRS is, instead, an exaggerated defense response to a noxious insult such as infections, trauma, surgery, or tumors [7,8]. Both ARDS and SIRS are associated with a dysregulated cytokine production that can cause a massive inflammatory cascade, leading to widespread organ failure.

An association has been widely demonstrated in COVID-19 between the presence of the so-called cytokine storm and disease severity, with a strong correlation linking high serum levels of pro-inflammatory mediators to a higher mortality risk [1,9,10,11]. A massive release of IL-10, IL-18, IL-33, IFNγ, IL-6, IL-1β, and TNF-α has been described upon SARS-CoV-2 infection; in particular, IFNγ, IL-6, IL-1β, and TNF-α are implicated in the onset of disease severity, likely by further promoting the activation of immune cells [12,13].

A recent study by Karki et al. conducted on bone marrow-derived macrophages from mice focused on the specific effects of several cytokines and found that only the combination of tumor necrosis factor-alpha (TNFα) and interferon-gamma (IFNγ) induced inflammatory cell death, with mechanisms that were dependent on the JAK/STAT1/IRF1 axis with inducible nitric oxide synthase (*NOS2*) induction [14]. The same study stated that *NOS2* is significantly upregulated in patients with severe and critical COVID-19 compared with healthy controls [14].

Nitric oxide (NO), generated from L-arginine by different isoforms of nitric oxide synthase (NOS), is a versatile free radical that exerts several physiological and pathological functions with both toxic and regulatory effects [15,16]. In endothelial cells, the production of an adequate release of NO through the activity of the constitutive eNOS is essential for vascular homeostasis, with beneficial effects on the regulation of vascular tone. On the contrary, NO produced by the inducible isoform iNOS can have both beneficial and detrimental effects [17,18] and is, indeed, endowed with antimicrobial and antitumor activities; however, a massive release of NO appears to be strongly implicated in the maintenance of chronic inflammation and can cause tissue damage through the generation of reactive nitric oxygen species (RNOS) such as peroxynitrites [16,19,20]. 

In macrophages, iNOS is highly expressed in response to cytokines, including TNFα, IFNγ, IL-6, and IL-1β and Toll-like receptor (TLR) ligands such as lipopolysaccharides (LPSs) and bacterial and viral components [21,22,23]. iNOS transcription is regulated by different transcription factors, including nuclear factor-κB (NF-κB), interferon regulatory factor-1 (IRF-1), and the signal transducer and activator of transcription-1 (STAT-1) phosphorylated dimers [24]. 

Recently, we showed that exposure to cytokines released by macrophages previously activated by the spike S1 protein of SARS-CoV-2 causes a growth arrest in alveolar A549 cells, with changes referable to the IFNγ-dependent induction of IFN-regulatory factor 1 (IRF-1) [25]. In the current study, we explored the role of cytokines produced by human macrophages exposed to spike S1 on iNOS expression in A549 cells, as well as the molecular pathways responsible for the effects observed.

## 2. Materials and Methods

### 2.1. Cell Models

The human wild-type A549 cell line (A549 WT, ab255450) and human IRF1 knockout A549 cell line (IRF1 KO, ab267042) were obtained from Abcam plc. (Prodotti Gianni S.r.l., Milan, Italy) and cultured in an RPMI1640 medium supplemented with 10% FBS and 1% penicillin/streptomycin at 37 °C in a humidified atmosphere with 5% CO_2_. For experiments, cells were seeded at a density of 2 × 10^5^ cells/mL in 24-well or 12-well culture plates.

### 2.2. Experimental Treatments

A549 cells were treated with conditioned media (CM) collected from monocyte-derived macrophages (MDM) and were obtained as already described [26,27]. CM were collected from MDM and incubated for 24 h in the absence (CM_cont) in the presence (CM_S1) of the 5 nM S1 subunit of SARS-CoV-2 spike recombinant protein (ARG70218; Arigo Biolaboratories by DBA-Italia S.r.l, Segrate, (MI), Italy) premixed with 2 µg/mL Polymyxin B, to exclude any possible contamination by lipopolysaccharide (LPS). The media obtained from the MDMs of 14 different donors were pooled and employed for the treatment of A549 cells. Alternatively, A549 were incubated with CM_cont, added with 50 ng/mL of IFNγ, IL-1β, TNFα, and IL-6 (R&D by Biotechne, Milan, Italy), and used either singly or in combination. Where indicated, cells were pre-treated with 1 µM of Baricitinib or 20 µM of Caffeic Acid Phenethyl Ester (CAPE) for 1 h before the addition of CM_S1; the inhibitor was left in a culture medium throughout the experiment.

### 2.3. RT-qPCR Analysis

Gene expression was analyzed using RT-qPCR, as previously described [28]. In total, 1 µg of RNA was reverse-transcribed to cDNA using the RevertAid First Strand cDNA Synthesis Kit (Thermo Fisher Scientific, Monza, Italy); qPCR was then performed on a StepOnePlus Real-Time PCR System (Thermo Fisher Scientific) by employing specific forward/reverse primer pairs (Table 1) and SYBR™ Green or a TaqMan Gene Expression Master Mix (Thermo Fisher Scientific). The amount of the gene of interest used upon treatment with the S1 protein was calculated using the 2^−∆∆Ct^ method [29] and expressed, relative to RPL15, as the fold change in control cells (=1).

### 2.4. Western Blot Analysis

Cell lysates obtained with an LDS sample buffer (Thermo Fisher Scientific) were employed for the analysis of protein expression as already described [30]. In total, 20 µg of proteins were separated on Bolt™ 4–12% Bis-Tris mini protein gel (Thermo Fisher Scientific) and transferred to PVDF membranes (Immobilon-P membrane, Thermo Fisher Scientific). Membranes were incubated for 1 h at RT in a blocking solution (4% non-fat dried milk in TBST, Tris-buffered saline solution + 0.5% Tween) and then overnight at 4 °C with an anti-IRF-1 or anti-iNOS rabbit polyclonal antibody (1:2000, Cell Signaling Technology, Euroclone, Pero, (MI), Italy) in TBST containing 5% BSA. The anti-vinculin mouse monoclonal antibody (1:2000, Merck, Milano, Italy) was used as a loading control. Horseradish peroxidase (HRP)-conjugated secondary antibodies (anti-rabbit and anti-mouse IgG, Cell Signaling Technology) were employed (1:10,000), and immunoreactivity was visualized using a SuperSignal™ West Pico PLUS Chemiluminescent HRP Substrate (Thermo Fisher Scientific). Western Blot images were captured with an iBright FL1500 Imaging System (Thermo Fisher Scientific) and analyzed with iBright Analysis Software (version 1.8.0).

### 2.5. Cytokine Analysis

Cytokines released by human macrophages and stimulated by spike S1 in the culture medium (CM_S1) were quantified with the Human Magnetic Luminex Screening Assay (R&D Systems, Bio-techne, Milano, Italy), according to the manufacturer’s instructions.

### 2.6. Cell Viability

Cell viability was assessed by employing the resazurin method [31]. A549 cells were incubated for 1 h with RPMI supplemented with 44 μM resazurin; after this period, the fluorescence of resorufin, derived from the transformation of resazurin by viable cells, was measured at 572 nm with a fluorimeter (EnSpire Multimode Plate Readers; PerkinElmer, Monza, Italy). 

### 2.7. Determination of Nitric Oxide Production

The production of nitric oxide (NO) was determined by means of a fluorimetric approach, addressing the production of the fluorescent molecule 1-(H)-naphtotriazole from 2,3-diaminonaphthalene (DAN) in an acid environment, as previously described [32]. In total, 100 μL of the cell medium was mixed with 20 μL of DAN (0.025 mg/mL in 0.31 M HCl). After 10 min at room temperature, 20 μL of 0.7 M NaOH was added, and fluorescence was read with the EnSpire^®^ Multimode Plate Reader (PerkinElmer, Milano, Italy). Nitrite production was expressed in nmoles/mL of the extracellular medium (μM).

### 2.8. Statistical Analysis

GraphPad Prism 9 (version 9.3.0, GraphPad Software, San Diego, CA, USA) was used for statistical analysis. *p* values were calculated with Ordinary One-way ANOVA for multiple comparisons or a One sample t-test, as specified in the legend of each Figure. *p* values < 0.05 were considered statistically significant.

### 2.9. Materials

R&D was the source of recombinant human cytokines: HEK293 expressed IFNγ; *E. coli*-derived IL-1β/IL-1F2 protein; HEK293 expressed TNFα; HEC293 expressed IL-6. Endotoxin-free fetal bovine serum was purchased from Thermo Fischer Scientific, while Baricitinib (Cayman Chemicals, Ann Arbor, MI, USA) was from Vinci-Biochem S.r.l., Firenze, Italy. Merck (Milano, Italy) was the source of CAPE, as well as all the other chemicals unless otherwise specified.

## 3. Results

In a recent contribution, we showed that the exposure of alveolar A549 cells to a conditioned medium (CM) obtained from human macrophages treated with spike S1 (CM_S1) led to the activation of epithelial cells, as highlighted by the increased production of many inflammatory mediators [26]. Here, by further addressing the immune-mediated effects of the spike protein, we evaluated the induction of iNOS in airway epithelial cells as a part of the inflammatory response. As shown in Figure 1, a time-dependent increase in the expression of *NOS2* mRNA was observed upon incubation with CM_S1, with a maximum effect detectable after 8 h of incubation. Consistently, the iNOS protein, already detectable after 4 h of incubation, was maximally expressed at 8 h and declined thereafter. 

As previously demonstrated, the conditioned medium from S1-treated human macrophages is rich in many cytokines and chemokines [25,33]. Among them, we show here that TNFα and IL-6 in the CM_S1 employed in this study reached concentrations of about 50 ng/mL, while IL-1β and IFNγ were about 1 ng/mL (Figure 2A). To address the role of these cytokines in the induction of iNOS in A549, cells were exposed to a mixture of these mediators at different doses. A dose-dependent increase in *NOS2* expression was evident with maximal stimulation when all the mediators were present at 50 ng/mL each (Figure 2B). To verify which of these mediators was mainly responsible for *NOS2* induction, we next incubated A549 WT in the presence of 50 ng/mL of cytokines, used either alone or in combination. The results presented in Figure 2C demonstrate that A549 WT IL-6 was ineffective, while IFNγ, TNFα, and IL-1β all slightly increased in their *NOS2* expression when employed alone. A more marked induction of *NOS2* was observed, instead, upon co-incubation with IFNγ and TNFα or, especially, IL-1β, but not IL-6, pointing to a synergism among the cytokines. Consistently, the simultaneous presence of IFNγ, TNFα, and IL-1β (cytomix) led *NOS2* expression to impressively high levels. 

The expression of *NOS2* is known to be modulated by IFN-regulatory factor 1 (IRF1); this transcription factor is, indeed, induced by IFNγ during the host immune response to viral infections [34]. Since we have recently observed an induction of IRF1 at both the gene and protein level upon the incubation of A549 cells with CM_S1 [25], we here explored the role of the transcription factor in the stimulation of iNOS expression by employing an IRF1 knockout A549 cell line (IRF1 KO). To this end, we first checked the absence of the protein in KO cells; as shown in Appendix A, IRF1 actually remained undetectable even when cells were treated with CM_S1. The cytokines responsible for *NOS2* induction in A549 WT cells were then tested in IRF1 KO. The results obtained, as shown in Figure 2C, indicated a pattern of expression similar to that of normal cells, although the effects observed were much lower in all the experimental conditions; consistently, the protein, readily detectable in WT cells at any time of incubation with cytomix, was much less expressed in IRF1 KO and barely detectable only after 8 h of incubation (Figure 2D). In line with this finding, nitrites and stable derivatives of nitric oxide were detectable after 24 h of incubation in WT but not in IRF1 KO cells (Figure 2E). 

The molecular mechanisms underlying iNOS induction were then investigated by employing baricitinib as an inhibitor of the JAK/STAT pathway [35] with CAPE as the inhibitor of NF-κB transcription factor [36]. As shown in Figure 3, baricitinib prevented the IFNγ-dependent induction of *NOS2* in WT cells, both when the cytokine was alone or combined with other mediators, while it had no effect on TNFα- or IL-1β-mediated stimulation. On the contrary, the inhibitory effect of CAPE in the same cells was complete for TNFα and IL-1β, either employed alone or together and only partial when IFNγ was present in the incubation medium. These findings were not ascribable to the cytotoxic effects of the experimental conditions adopted; indeed, no change in cell viability was observed for short-term treatments. A significant, albeit modest, cell loss was detectable only after 24 h incubations (Appendix A). The pattern of inhibition in IRF1 KO cells was similar to that of WT cells with respect to baricitinib; in these cells, however, the inhibitory effect of CAPE was complete on TNFα and IL-1β and also when IFNγ was present. Overall, these results sustain the involvement of IRF1 and NF-κB transcription factors in the induction of *NOS2* by the cytokines, with a prominent role for IRF1 when IFNγ was present.

When further addressing the role of this latter transcription factor in the induction of iNOS by CM_S1, we observed that the incubation of IRF1 KO cells with a conditioned medium stimulated *NOS2* gene expression only at very modest levels compared to WT cells (Figure 4, left panel). Moreover, the presence of baricitinib completely prevented the CM_S1-dependent induction of *NOS2* mRNA in both WT and IRF1 KO cells; conversely, the efficacy of CAPE in limiting NOS induction, although evident, remained incomplete in both cell models. In line with mRNA data, the expression of the iNOS protein in WT cells was completely suppressed by baricitinib at any time, while a band was still detectable, although fainter, in the presence of CAPE (right panel); as far as IRF1 KO cells are concerned, a weak band was barely detectable after only 8 h of incubation and was abolished by both baricitinib and CAPE.

## 4. Discussion

The deregulation of nitric oxide (NO) metabolism has been recently related to the pathogenesis of COVID-19 and to the development of ARDS in severe patients [37]. The precise role of this molecule, however, has still to be fully elucidated. Focusing on the roles of the constitutive eNOS and inducible iNOS isoforms, it has been proposed that hyperinflammation leads to endothelial dysfunction and to the impairment of eNOS-derived NO production, which, in turn, causes systemic alterations, especially in the vascular system. At the same time, iNOS activity and NO production are enhanced in the effort to fight the virus, but when deregulated, they can contribute to lung injury and ARDS progression [37]. Evidence in the literature has reported that, at least in the first wave of COVID-19 patients, serum iNOS levels were increased, and iNOS was proposed to be predictive for the COVID-19 outcome [38]; consistently, Karki et al. found that *NOS2* expression was significantly upregulated in patients with severe and critical COVID-19 compared to healthy controls [14]. 

Here, we show that the conditioned medium of macrophages exposed to spike S1 of SARS-CoV-2 potently induces iNOS expression in alveolar epithelial A549 cells, which is likely due to the simultaneous presence of IFNγ, IL-1β, and TNFα. These cytokines, indeed, slightly stimulate *NOS2* expression when employed alone, but, when combined, IFNγ acts synergistically with TNFα and, even more, with IL-1β; when employed all together, they stimulate *NOS2* expression to very high values, which is comparable to those obtained with conditioned medium. Clearly, we cannot exclude the fact that other mediators released by S1-activated macrophages in the conditioned medium are involved in *NOS2* induction in alveolar epithelial cells. Actually, it is conceivable that the incubation of macrophages with a spike leads to the release of many other molecules besides inflammatory cytokines; among them, for example, we have recently described a massive secretion of chemokines including IL-8, IP-10, and RANTES [26,33]. However, our findings presented here clearly ascribe to IFNγ, IL-1β, and TNFα an undeniable role in the CM_S1-dependent stimulation of iNOS expression and activity. Similarly, the up-regulation of iNOS expression in intestinal Bowel Diseases (IBD) is supposed to involve proinflammatory cytokines since a positive correlation is established between NO production and increased levels of TNF-α, IFNγ, IL-17, IL-12, and IL-6 in IBD patients [39]. Under our experimental conditions, instead, IL-6, despite its well-recognized correlation with COVID-19 severity [40], was completely ineffective, excluding the role of this cytokine in *NOS2* induction. 

iNOS was first isolated in murine macrophages and subsequently found in many other cell types, including human alveolar epithelial cells [41,42,43]. Compared with other species, the induction of iNOS in human cells was limited, and substantial differences existed in the transcriptional regulation of the murine and human iNOS genes [44,45]. Under our experimental conditions, the protein was clearly detectable; however, the production of nitric oxide appeared limited. On the other hand, while it is noteworthy that stimulated murine macrophages produced huge amounts of NO [46], human macrophages failed to produce sufficient amounts of NO under multiple different induction conditions [47], possibly for a limited biosynthesis of the essential co-factor H4B [48].

To date, studies addressing the molecular pathways underlying *NOS2* induction have mainly been performed in murine macrophages and point to the involvement of the JAK/STAT1/IRF1 pathway. Indeed, in RAW 264.7 cells, the induction of the *NOS2* gene by gliadin and IFNγ has been ascribed to IRF1, STAT1α, and NF-κB transcription factors [49]. Other results from theoretical model studies have indicated that incubation with IFNγ and the subsequent activation of IRF1 is essential for the stimulation of iNOS by LPS, with priming by IFNγ appearing more significant than TNFα [24]. Recently, in murine bone marrow-derived macrophages, the synergism of TNFα and IFNγ was found to engage the JAK/STAT1 axis to induce IRF1 expression and NO production [14]. In humans, a single-cell transcriptional study performed in PBMCs from patients with COVID-19 suggested that the IFNα and IFNγ function in T cells and dendritic cells promote disease severity by activating STAT1 [50]. In line with this evidence from the literature, our results obtained in IRF1-deficient A549 cells demonstrate that the main route for iNOS induction by conditioned medium of S1-treated macrophages or cytomix involves the activation of the IRF1 transcription factor through the stimulation of the JAK/STAT pathway; a small amount of *NOS2* transcription is, however, IRF1-independent and likely relies on NF-kB. Given the efficacy of baricitinib in preventing the induction of iNOS in IRF1 KO cells, we can hypothesize that NF-κB-mediated effects are also under the control of the JAK/STAT axis; consistently, a transcriptional synergism between NF-κB and STAT1 in the regulation of inflammatory gene expression has been previously reported [51,52]. 

Overall, these observations allow a model to be drawn for IFNγ-, IL-1β, and TNFα-mediated signal transduction pathways that lead to iNOS induction upon incubation with CM_S1 (Figure 5). According to this model, the cytokines produced by macrophages and activated by the spike S1 protein of SARS-CoV-2, in particular IFNγ, IL-1β, and TNFα, induce *NOS2* expression in alveolar epithelial cells with a mechanism that is mainly, although not exclusively, dependent upon the JAK/STAT/IRF1 pathway. These findings gain particular relevance when considering the therapeutic use of baricitinib, already approved for the treatment of severe COVID-19. Indeed, in addition to shaping a patient’s immune response [53] and suppressing cytokine hyperproduction by targeting JAK/STAT [54], the use of this drug could also be beneficial for the prevention of iNOS expression.

## Figures and Tables

**Figure 1 biomedicines-11-02699-f001:**
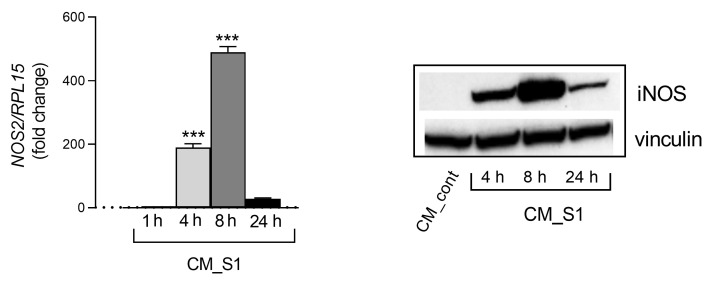
A549 WT were incubated with a conditioned medium (CM) from monocyte-derived macrophages (MDM) obtained through the incubation of MDM for 24 h in the absence (CM_cont) or in the presence of 5 nM S1 (CM_S1). At the times indicated, the expression of *NOS2* was measured by means of RT-qPCR (left panel) and calculated relatively to CM_cont (=1; dotted line) upon normalization for the housekeeping gene RPL15. Bars were the means ± SEM of three experiments, each performed in duplicate. At the same time, the amount of the iNOS protein was assessed by means of Western Blot analysis (right panel); representative blots are shown for three different experiments. *** *p* < 0.001 vs. CM_cont with Ordinary One-way ANOVA.

**Figure 2 biomedicines-11-02699-f002:**
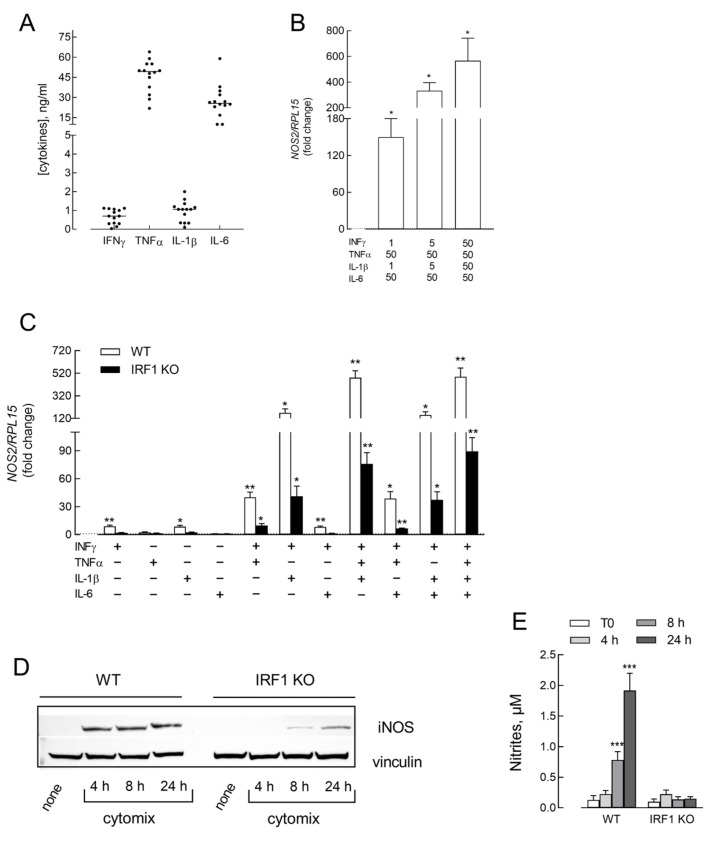
(**A**) Cytokines released by S1-treated macrophages from different donors were quantified as described in the Methods section. Data are the means ± SEM of 14 determinations, each performed in duplicate. (**B**) A549 WT were incubated with the indicated concentrations of cytokines. After 6 h, the expression of *NOS2* was measured by means of RT-qPCR and calculated relatively to CM_cont (=1; dotted line) upon normalization for the housekeeping gene RPL15. * *p* < 0.05 vs. CM_cont with One sample t-test. (**C**) A549 WT and IRF1 KO were incubated in the presence of 50 ng/mL of the indicated cytokines. After 6 h, the expression of *NOS2* was measured, as described in (**B**). Bars are means ± SEM of four experiments, each performed in duplicate. * *p* < 0.05; ** *p* < 0.01 vs. CM_cont with One sample t-test. (**D**) A549 WT and IRF1 KO were incubated in the presence of cytomix (IFNγ + TNFα + IL-1β, 50 ng/mL each). At the times indicated, the expression of the iNOS protein was assessed by means of Western Blot analysis; representative blots are shown for three different experiments. (**E**) A549 WT and IRF1 KO were incubated in the presence of cytomix. At the indicated times, the number of nitrites in the incubation medium was determined, as described in the Methods section. Bars are the means ± SEM of three independent experiments. *** *p* < 0.001 vs. T0 with Ordinary One-way ANOVA.

**Figure 3 biomedicines-11-02699-f003:**
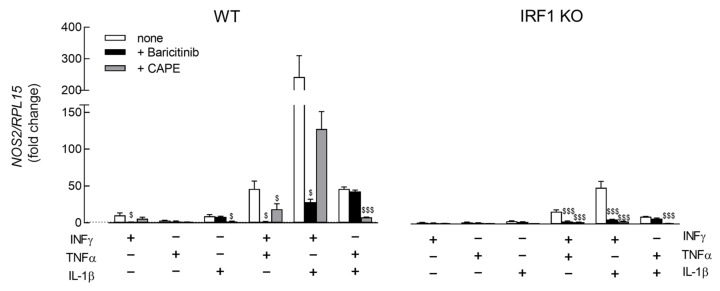
A549 WT and IRF1 KO were incubated in the presence of 50 ng/mL of the indicated cytokines in the absence or the presence of 1 µM baricitinib or 20 µM CAPE. After 6 h, the expression of *NOS2* was measured by means of RT-qPCR and calculated relatively to CM_cont (=1; dotted line) upon normalization for the housekeeping gene RPL15. Bars are the means ± SEM of three experiments, each performed in duplicate. ^$^
*p* < 0.05, ^$$$^
*p* < 0.001 vs. none with Ordinary One-way ANOVA.

**Figure 4 biomedicines-11-02699-f004:**
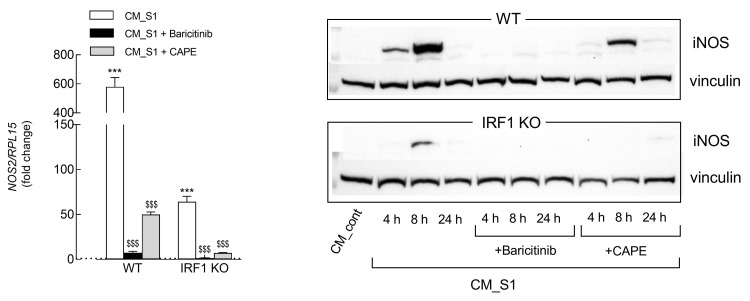
A549 WT and IRF1 KO were incubated in CM_cont or CM_S1 in the absence or the presence of 1 µM of baricitinib or 20 µM CAPE. Left panel. After 4 h, the expression of *NOS2* was measured by means of RT-qPCR and calculated relatively to CM_cont (=1, dotted line) upon normalization for the housekeeping gene RPL15. Bars are the means ± SEM of three experiments, each performed in duplicate. Right panel. At the indicated times, the amount of the iNOS protein was assessed by means of Western Blot analysis; representative blots are shown in three different experiments. *** *p* < 0.001 vs. CM_cont; ^$$$^
*p* < 0.001 vs. CM_S1 with Ordinary One-way ANOVA.

**Figure 5 biomedicines-11-02699-f005:**
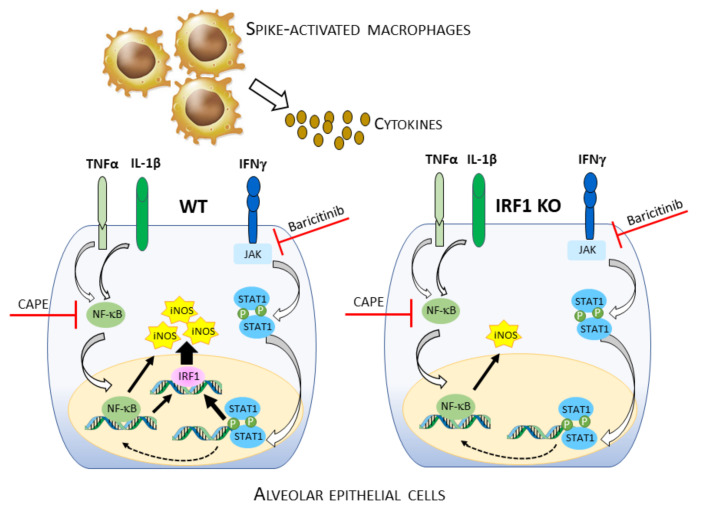
Proposed model of iNOS expression in human A549 epithelial cells.

**Table 1 biomedicines-11-02699-t001:** Sequence of primer pairs employed for RT-qPCR analysis.

Gene/Protein	Forward Primer	Reverse Primer
*RPL15*/RPL15	GCAGCCATCAGGTAAGCCAAG	AGCGGACCCTCAGAAGAAAGC
*IRF1*/IRF1	CTGTGCGAGTGTACCGGATG	ATCCCCACATGACTTCCTCTT
*NOS2*/iNOS	CACGCTCGCCTTCAAGTTC	AGGCACTAATGTAGGACCCAG

## Data Availability

All data are available at https://osf.io/fp7kg/files/osfstorage/650d5a8a5a0a812a94044a51 (accessed on 22 September 2023).

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
