# Peer review of "Cytokine-Induced iNOS in A549 Alveolar Epithelial Cells: A Potential Role in COVID-19 Lung Pathology"

_biomedicines, 2023, doi:10.3390/biomedicines11102699_

Round 1

Reviewer 1 Report

Overall, the author continues work based on a previous publication to address signaling pathways in the proinflammatory response making this paper relevant to novel insight on COVID-19 inflammatory pathogenesis.  There are a few corrections needed for clarity of the overall message, they are listed below

1)In figure 4, WT control at 24 hours does not show a band for iNOS, which is inconsistent with figure 1 control blood showing the same experimental design with CM_S1 media making this blot seem unreliable. Results should be consistent.

2) Author states that “findings sustain the therapeutic efficacy of baricitinib in COVID-19”, this statement is premature with the results only having been tested invitro,  no animal studies or clinical supporting evidence (plasma/biopsy results from COVID-19 patients in a clinical setting)

3) The author uses a dosage of 50ng of cytokines with no dose curve or justification for the chosen dosage? Author should show data to  justify the chosen dosage?

4) There is no cell viability data/ cell quantification to compare the effects of the spike protein or cytokines on cell health/viability, this is needed in order to confirm that the reduction in signaling is actually caused by inhibitory effect of baricitinib or CAPE and not by cell death.  Cell viability should be presented.

Author Response

We would like to thank the Reviewer for his/her collaborative criticisms; here our response to his/her comments:

  • Regarding the first question, we do not find any inconsistency between the results in Figures 1 and 4. In fact, the band corresponding to iNOS in cells treated for 24 h with CM_S1 is weak in Figure 1 and almost undetectable in Figure 4, but this was quite predictable, since the results were obtained in two different experiments. The overall meaning is the same for the two Figures, i.e. that the induction of iNOS by CM_S1 in A549 cells is progressive and transient: the protein, absent in untreated cells, reaches maximum expression after approximately 8 h of incubation, and then decreases again within 24 h.

  • We agree with the Reviewer that the final sentence was perhaps too assertive; we have modified it and now we hope it is more appropriate.

  • The choice of cytokines concentration in cytomix is now explained in the revised version of the ms.; to this end, we have now added two panels to Figure 2.

  • As far as cell viability under our experimental conditions is concerned, we now show the data in the new Figure S2, as suggested by the Reviewer. To this concern, however, we have already demonstrated that conditioned medium from S1-treated monocytes causes a cell cycle arrest, without any sign of cytotoxicity (Barilli et al. Biomedicines 2022; 10:3085); similarly, we are now preparing a paper where we demonstrate that cytomix blocks cell proliferation, mainly due to the presence of IFNγ, without signs of cytotoxicity or cell death. These effects, however, appear late (at least after 24 h), while the experimental conditions adopted in the present paper last within 24 h. As for the inhibitors, baricitinib exerts no cytotoxic effect, as already demonstrated (Barilli et al. Biomedicines 2022; 10:3085), while CAPE causes a slight cell loss after 24 h of incubation, when, however, we no longer observe any inhibitory effects on iNOS expression. In this regard, we would like to thank the Reviewer because, when reading his/her comment, we realized we had made a typo: the unit for CAPE concentration was µM instead of µg/µl and has been corrected accordingly.

Reviewer 2 Report

The article explores the potential role of cytokine-induced iNOS in COVID-19 lung pathology. The authors suggest that iNOS expression may contribute to the development of acute respiratory distress syndrome (ARDS) in COVID-19 patients. The authors suggest that cytokines such as IFN-γ and TNF-α can induce iNOS expression in alveolar cells. They propose that this may contribute to the development of ARDS in COVID-19 patients.

The authors suggest that inhibiting JAK/STAT and NF-kB signaling pathways may be a potential therapeutic strategy for COVID-19. They propose that this may help to reduce cytokine-induced iNOS expression and prevent the development of ARDS. However, they note that further research is needed to explore the potential benefits and risks of this approach.  

The article is suitable and accurate, without many pretensions, but it provides necessary information regarding the ARDS and opens the door to future research

Author Response

We would like to thank the Reviewer for the attention he/she paid to our manuscript and for his/her comments.

Reviewer 3 Report

The authors attempted to model the inflammation-driven iNOS upregulation (by macrophage-conditioned medium and recombinant cytokines) and the subsequent NO production in A549 cells. The roles of specific transcription factors NF-kB, IRF1, STAT1 in iNOS induction were investigated using inhibitors or KO models. While the idea could be interesting, the experimental setup has several important drawbacks.

Firstly, the recombinant Spike S1 used in the study was derived from E. coli. Thus, the eukaryotic glycosylation patterns are lacking in this protein. While it may cause macrophage activation, this is not necessarily because it is viral. As glycosylation plays an important role specifically for the S protein (https://doi.org/10.1038/s41392-021-00809-8) the similarity of this model to SARS-CoV-2 infection in vivo is rather questionable and requires a CHO- or HEK-derived viral protein.

Second, the four cytokines used in the study are also supposedly recombinant proteins, however, their cell source is unclear. Similarly, their choice and concentrations used are not justified. Supposedly, they are the effectors induced by Spike S1 in macrophages in the previous studies, however, the text would benefit from more explanation there.

The smaller points include:

Third, was the Spike S1 protein eliminated from the macrophage-conditioned medium before it was transferred to A549 cells (Line 80)?

Fourth, the use of TF inhibitors is important to confirm their role in inflammatory response.  However, in view of A549 cells having distinct basal levels of ABC-transporters that may cause a MDR-resistant phenotype in them, did Baricitinib and CAPE remain in the cells? And was not the lack of effect (e.g. with CAPE) due to the lack of inhibitor (Line 216)?

Fifth, it is not clear from the text or from the reference, how the authors distinguished 1 ug cDNA from RNA after reverse transcription (Line 93).

In summary, while the manuscript is interesting and well-written, there are important issues that have to be addressed before publication.

Author Response

First, we would like to thank the Reviewer for his/her comments; here we add a point-by-point answer to his/her concerns:

  • We are aware that differences exist between the effects of recombinant S1 protein in vitro and viral infection in vivo; similarly, we know that the different efficacy of recombinant S1 isolated from E. Coli and mammalian cells is still a matter of debate. However, among the numerous contributions in the literature addressing the effects of S1 protein on immune cells (including monocytes, PBMC and THP-1 cells) the pattern of cytokine secretion is roughly comparable, regardless of the origin of the spike used, and adequately reflects the cytokine profile observed in the serum of COVID-19 patients. Furthermore, to date, the specific impact of glycosylation on the mechanism of action of SARS-CoV-2 is still controversial and it has been proposed that the molecular structure of the protein, more than the presence of glycans, may be relevant for spike interaction with receptors (Shirato & Kizaki, Helyon, 2021, 7:e06187).
  • The source of recombinant cytokines is now detailed in Materials, as requested by the Reviewer; also the reason for choosing the concentration is now better explained through the addition of new Panels A and B in Figure 2 of the revised ms.
  • As far as CM_S1 is concerned, we confirm that spike protein was not removed before the incubation of A549; however, we have already demonstrated in a previous contribution that 10 nM S1 alone has no effect on A549 (Barilli et al., Biomedicines 2022, 10(3), 618)
  • As for the efficacy of TF inhibitors, results in Figure 3 show that both baricitinib and CAPE are able to prevent NOS2 induction under specific conditions: baricitinib inhibits all IFNg-mediated effects, while CAPE is effective toward TNFα and IL-1β.
  • We agree with the Reviewer that the sentence about reverse transcription was unclear, so we now modified the text.

Round 2

Reviewer 1 Report

The author addressed our concerns for the paper and revised the western blot in figure 4 as well as performed a cell viability assay in supplemental figures to demonstrate that the results shown in the study are not due to cellular damage, but rather the effect of the study. I recommend publication of this article in its current state.